# An inverse dielectric mixing model at 50MHz that considers soil organic carbon

Chang-Hwan Park[1,2], Aaron Berg[3], Michael H. Cosh[4], Andreas Colliander[5], Andreas Behrendt[6], Hida Manns[3], Jinkyu Hong[7], Johan Lee[1], Runze Zhang[8] and Volker Wulfmeyer[6]

[1]National Institute of Meteorological Sciences, Earth System Research Division, Korea Meteorological Administration
[2]Department of Civil Systems Engineering, Ajou University, Tashkent, Uzbekistan
[3]Department of Geography, Environment and Geomatics, University of Guelph, Guelph, ON N1G 2W1, Canada
[4]United States Department of Agriculture, Agricultural Research Service, Hydrology and Remote Sensing Laboratory, Beltsville, MD 20705, USA
[5]Jet Propulsion Laboratory, California Institute of Technology, Pasadena, CA 91109, USA
[6]Institute of Physics and Meteorology, University of Hohenheim, Stuttgart 70599, Germany
[7]Ecosystem-Atmosphere Process Lab., Dep. of Atmospheric Science, Yonsei Univ., Seoul, 03722 Republic of Korea
[8]Department of Engineering Systems and Environment, University of Virginia, Charlottesville, VA 22904, USA

*Correspondence to*: Chang-Hwan Park (ecomm77@gmail.com or cpark@ajou.ac.kr)

**Abstract.** The prevalent soil moisture probe algorithms are based on a polynomial function that does not account for the variability in soil organic matter. Users are expected to choose a model before application: either a model for mineral soil or a model for organic soil. Both approaches inevitably suffer from limitations with respect to estimating the volumetric soil water content in soils having a wide range of organic matter content. In this study, we propose a new algorithm based on the idea that the amount of soil organic matter (SOM) is related to major uncertainties in the in-situ soil moisture data obtained using soil probe instruments. To test this theory, we derived a multiphase inversion algorithm from a physically-based dielectric mixing model capable of using the SOM amount, performed a selection process from the multiphase model outcomes, and tested whether this new approach improves the accuracy of soil moisture (SM) data probes. The validation of the proposed new soil probe algorithm was performed using both gravimetric and dielectric data from the Soil Moisture Active Passive Validation Experiment in 2012 (SMAPVEX12). The new algorithm is more accurate than the previous soil-probe algorithm, resulting in a slightly improved correlation (0.824 $\rightarrow$ 0.848), 12 % lower root mean square error (RMSE; 0.0824 $\rightarrow$ 0.0727 cm$^3$·cm$^{-3}$), and 95 % less bias (-0.0042 $\rightarrow$ 0.0001 cm$^3$·cm-$^3$). These results suggest that applying the new dielectric mixing model together with global SOM estimates will result in more reliable soil moisture reference data for weather and climate models and satellite validation.

## 1. Introduction

Soil moisture (SM) plays a critical role in weather and climate by affecting atmospheric variables via latent and sensible heat exchange. For example, near-surface air temperature can be affected by the evapotranspiration of surface and root zone soil moisture. Therefore, its correlation with the near-surface temperature is usually considered an effective indicator of the coupling strength between the land surface

and the atmosphere (Seneviratne et al., 2006; Koster et al., 2009; Seneviratne et al., 2010; Jaeger and Seneviratne, 2011; Seneviratne et al., 2013; Hirschi et al., 2014; Whan et al., 2015). In particular, soil moisture anomalies in a dry regime have been reported as the main cause of strong land-atmosphere coupling, which can trigger drought and heat waves (Fischer et al., 2007; Zampieri et al., 2009; Guillod et al., 2015; Hauser et al., 2016; Hirschi et al., 2011; Miralles et al., 2011; Taylor et al., 2012; Mueller and Seneviratne, 2012; Seo et al., 2019). Soil moisture also influences precipitation formation and storm tracks by coupling with the atmosphere (Koster et al., 2004; Taylor et al., 2012; Guillod et al., 2015; Santanello et al., 2018, 2019; Zhang et al., 2019). Consequently, inaccurate SM information in the land-surface-model hinders accurate predictions of extreme climate and weather because of unrealistic land-atmosphere interactions that result from uncertainties in air temperature, moisture, dynamics, cloud formation and precipitation.

High-quality in-situ soil moisture data are an important reference for evaluating climate models (Yuan and Quiring, 2017; Zhuo et al., 2019) and remote-sensed SM data (Entekhabi et al., 2010; Kerr et al., 2010) . However, it is not practically possible to perform in-situ SM measurements with high spatial and temporal coverage. Soil moisture networks based on cosmic ray neutron probes might be more manageable for long-term operation, however this approach - and associated networks are not established, globally, as dielectric based approach. A practical alternative is to employ a portable soil probe that is calibrated using locally measured soil moisture. In particular, portable dielectric sensors make use of the relationship between the dielectric constant and volumetric soil water content. However, such retrieval of the volumetric soil water content from dielectric measurements does not account for soil organic matter (SOM) and saturation conditions. A few studies have reported the relationship between the dielectric constant and the volumetric soil water content in organic soils (Topp et al., 1980; Roth et al., 1992; Bircher et al., 2012). However, the calibration functions derived from these studies have limitations for global-scale applications because they were developed using only a few specific sites and/or applicable only for the sites with a limited range of organic matter content. For the purpose of a global soil moisture probe observing system, using an inversion method of the existing physical dielectric mixing model can be a great alternative approach to incorporate the variability of organic matter into the probe algorithm beyond the current empirical probe models.

With this background, this work provides a pathway for a physical model to consider soil organic matter. We developed an inverse dielectric mixing model for mineral soil derived from Park et al. (2019, 2017) to obtain more accurate volumetric soil moisture estimates from the dielectric constant. The proposed model reflects the damping effect and simulates the supersaturation of soil moisture over soil porosity (when soil moisture occupied larger than porosity of dry compacted soil in the unit volume causing light weight clay swelling or starting existence of standing water or starting surface runoff due to the precipitation accumulation over soil surface faster than infiltration) so that we can capture the standing water and surface runoff during flood events, which has not been studied in other prevalent dielectric mixing models

The most recent high resolution SOM map (Hengl et al., 2014; Batjes, 2016) is only available as a static variable for the land model; therefore, the realism of the parameterization for surface runoff, infiltration, evapotranspiration, and soil respiration is limited. Therefore, it would be important to obtain a spatially and temporally varying SOC map from satellite measurements. SMAP has the potential to provide an

unprecedented and unique benefit to solve various challenges in deriving such maps regardless of the relatively coarse resolution of the SMAP radiometer measurements because of the following three reasons: 1) microwaves can detect SOC underneath vegetation, which other shorter wave sensors cannot perceive; 2) the temporally varying OC evolution obtained even from low-resolution satellite image will be helpful in various modelling and observation studies, and 3) the limitation of the low-resolution issue can be overcome by recent downscaling approaches, such as machine learning methods, that can utilize a synergy with other ground, spaceborne and satellite data. Consequently, the other aim of this study is to provide a foundation for global SOM estimation using observations from a satellite, such as Soil Moisture Active Passive (SMAP), by developing a dielectric mixing model based on accurate in-situ SOM and gravimetric soil moisture.

The remainder of this paper is organized as follows: Section 2 introduces the inversion approach of dielectric mixing model to estimate soil moisture from organic-rich mineral soil using the probe. The data used in this study are described in Sect. 3. In Sect. 4, we evaluate the results using the soil moisture measured during SMAPVEX12. Finally, a summary and discussion for further applications are provided in Sect. 5.

## 2. Method

The dielectric constant indicates a polarizability of materials at a certain wavelength. The dipole structure of water molecules is highly sensitive to a microwave electric field with very high dielectric constant (approximately 80). On the other hand, the dielectric constant of mineral soil at microwave electric fields is rarely reacting, having only low value from 3 to 5. Therefore, an instrument which can measure the effective dielectric constant of soil medium such as Stevens Hydraprobe can provide an accurate estimate of water amount within soil (Jackson et al., 1982; Schmugge, 1983; Stafford, 1988). Also, from space, microwave satellite such as SMAP (Soil Moisture Active Passive) (Entekhabi et al., 2010), SMOS (Soil Moisture and Ocean Salinity) (Wigneron et al., 2007) and AMSR-E (Advanced Microwave Scanning Radiometer for EOS) can effectively estimate soil moisture from the measured brightness temperature by relating the effective dielectric constant of land surface.

For the application of portable soil moisture probes, the in-situ soil moisture data are provided based on the empirical relationship between the measured dielectric constant and the volumetric soil moisture (Seyfried and Murdock, 2004; Bell et al., 2013) using the following equation:

$$w = 0.0838\sqrt{\varepsilon_{\mathrm{obs}}} - 0.0846 \tag{1}$$

where, $\varepsilon_{obs}$ is the real part of the dielectric constant measured with the soil probe and $w$ is the estimation of the volumetric soil moisture ($cm^3 \cdot cm^{-3}$). As apparent in Eq. (1), the dependence of $\varepsilon_{\mathrm{obs}}$ on SOM was not considered in the estimation of $w$.

To consider the SOM, we first derive Eqs. (2)–(4), based on Park et al. (2019).

If the observed real part of the dielectric constant measured with the soil probe is smaller than the real part of the dielectric constant at the wilting point, $\varepsilon_{\mathrm{obs}} < \varepsilon_{\mathrm{wp}}$, we obtain:

for $w < w_{\mathrm{wp}}$

$$w = a\big((\varepsilon_{\text{obs}} - 1)H^{-1} + 1\big) + b \qquad (2)$$

where,

$$a = 1/(\varepsilon_{\text{bound}} - \varepsilon_{\text{air}})$$

$$b = -\frac{(1-p)\varepsilon_{\text{soil}} + p\varepsilon_{\text{air}}}{\varepsilon_{\text{bound}} - \varepsilon_{\text{air}}}$$

where, $H$ is the damping factor (0.8), $\varepsilon_{\text{bound}}$, is the dielectric constant for bound water, $\varepsilon_{\text{free}}$, dielectric constant for free water, $\varepsilon_{\text{air}}$ is the dielectric constant for air (1).

If the observed real part of the dielectric constant measured with the soil probe is larger than the real part of the dielectric constant at the wilting point and still smaller than the saturation point, $\varepsilon_{\text{wp}} < \varepsilon_{\text{obs}} < \varepsilon_{\text{p}}$, we get:

for $w_{\text{wp}} < w < p$

$$w = \frac{-b + \sqrt{b^2 - 4a(c - (\varepsilon_{\text{obs}} - 1)H^{-1} - 1)}}{2a} \qquad (3)$$

where,

$$a = \frac{\varepsilon_{\text{free}} - \varepsilon_{\text{bound}}}{p - w_{\text{wp}}}$$

$$b = \frac{p\varepsilon_{\text{bound}} - w_{\text{wp}}\varepsilon_{\text{free}}}{p - w_{\text{wp}}} - \varepsilon_{\text{air}}$$

$$c = (1-p)\varepsilon_{\text{soil}} + p\varepsilon_{\text{air}}$$

Finally, for $\varepsilon_{\text{obs}} > \varepsilon_{\text{p}}$, we get:

for $p < w$

$$w = a\big((\varepsilon_{\text{obs}} - 1)H^{-1} + 1\big) + b \qquad (4)$$

$$a = \frac{1}{\varepsilon_{\text{free}} - \varepsilon_{\text{soil}}}$$

$$b = -\frac{\varepsilon_{\text{soil}}}{\varepsilon_{\text{free}} - \varepsilon_{\text{soil}}}$$

According to Debye Relaxation, the dielectric constant of free water at less than 2GHz frequency has a constant value of approximately 80. However, in the field measurements (Curtis, John O. et al., 1995; Ishida, 2000; Mironov et al., 2013; Fal et al., 2016) it is found that in clay-rich soil, the real part of the dielectric constant increases at lower frequencies, which occurs by the clay-ion-complex interaction (Kelleners et al., 2005). Therefore, in this study for 50 MHz, the clay content and the real part of the dielectric constant at 1.4GHz are empirically considered in the dielectric constant not only for free, but also, for bound water Eq.(5,6).

$$\varepsilon_{free} = \varepsilon_{free_{1.4GHz}} + 65 \cdot v_{clay} \qquad (5)$$

$$\varepsilon_{bound} = \varepsilon_{bound_{1.4GHz}} + 5 \cdot v_{clay} \tag{6}$$

Also, we proposed the formulation of the dielectric constant for the dried organic-rich mineral soil at 50 MHz, as shown in Eq. (7).

$$\varepsilon_{soil} = \left(\varepsilon_{clay} \cdot v_{clay} + \varepsilon_{sand} \cdot v_{sand} + \varepsilon_{silt} \cdot v_{silt}\right)(1 - v_{SOM}) + \varepsilon_{SOM} \cdot v_{SOM}$$

$$\tag{7}$$

where, $\varepsilon_{\text{free 1.4GHz}}$ and $\varepsilon_{\text{bound 1.4GHz}}$ are the dielectric constant for free and bound water at 1.4GHz, respectively and $v_{clay}$, $v_{silt}$ and $v_{sand}$ are the volumetric ratios ($cm^3 \cdot cm^{-3}$) for clay, silt and sand, respectively.

The bulk density for organic soils can be computed with pure mineral and organic matter densities (Federer et al., 1993) or be expressed with their total volume and mass of these component (Liu et al., 2013; Jin et al., 2017). By relating these two formulas, we can derive the following volumetric ratio of organic matter ($v_{SOM}$, $cm^3 \cdot cm^{-3}$) (see appendix A for more details):

$$v_{SOM} = \left(\left(\frac{1}{SOM} - 1\right)\frac{BD_{SOM}}{BD_{MI}} + 1\right)^{-1}$$

$$\tag{8}$$

where,

$$SOM \text{ [kg/kg]} = f_{oc} \cdot \frac{OC \text{ [g/kg]}}{1000} \tag{9}$$

$$BD = 0.071 + 1.322 \cdot exp(-0.0071 \cdot OC) \tag{10}$$

SOM is expressed as organic carbon (OC) in the majority of global soil maps (Hugelius et al., 2013; Hengl et al., 2014, 2017, "Harmonized world soil database v1.2 | FAO SOILS PORTAL," 2020) as well as in the published units in the SMAPVEX 12 study (Manns and Berg, 2014). Organic carbon is the major component of SOM, and in order to convert OC to SOM, the conversion factor ($f_{oc}$) of 1.8 was used in Eq (9). The conventional OC-to-SOM conversion factor was proposed to be 1.724 by (Waksman and Stevens, 1930; Stenberg et al., 2010). However, it has been reported that the OC-to-SOM conversion factor can vary from 1.25 to 2.5, and the conventional value of 1.724 tends to overestimate the OC, as reported by Pribyl (2010). Instead of 1.724, 1.8 is a more appropriate value for a wide range of OC, as supported by various studies (Broadbent, 1953; Ranney, 1969; Manns and Berg, 2014). Therefore, in this study, we applied 1.8 for the conversion factor $f_{oc}$ in Eq. (9). If a further effort in mapping conversion factors in global scale is made in a future study, the probe sensor algorithm might benefit in the improvement of its accuracy for soil moisture estimation in organic and peat soils.

By applying Eq.(10) (Hossain et al., 2015), $BD_{MI}$ (bulk density of "pure" mineral matter) and $BD_{SOM}$ (bulk density of "pure" organic matter) in Eq.(8) are computed as 1.393 $g \cdot cm^{-3}$ with 0% OC (0g OC per 1kg soil) and 0.097 $g \cdot cm^{-3}$ with 56% OC (560g OC per 1kg soil) converted from 100% SOM with the conversion factor 1.8 by Eq.(9), respectively.

In a previous study, Eq. (11) was proposed as the wilting point, which is a function of SOM ($kg \cdot kg^{-1}$) with the slope parameter of SOM modified from 0.786 to 0.6 (Park et al., 2019). In our study the porosity

is suggested as a power law function according to the SOM variable, as shown in Eq. (12) (please see the result of simulation in all SOM and clay regions in Figure B1 in Appendix B).

$$w_{wp} = 0.02982 + 0.089 \cdot v_{clay} + 0.65 \cdot \text{SOM} \tag{11}$$
$$p = 0.194 + 0.26 \cdot v_{clay} + 0.5 \cdot \text{SOM}^{0.5} \tag{12}$$

By applying Eqs. (11) and (12), which require Eq. (8), Eqs. (2–4) can be used to compose the inverse
dielectric mixing model for organic-rich mineral soil (IDO). A detailed description of the parameters used in the algorithm is provided in Table 1. Previous studies (K. E. Saxton et al., 1986; Vereecken et al., 1989; Schaap et al., 1998, 2001; Chadburn et al., 2015) showed that greater SOM values increase the wilting point and porosity as proposed in equations (11-12). This relationship between the organic matter and the soil parameters might become more complex in organic rich soil if the type of OC is also important and
the dominancy for the OC type change in high SOC region. Such a complex relationship should be considered by including detail classification of SOM as sapric, hemic and fibric based on previous study (Verry et al., 2011).

**Table 1. Required physical properties to inverse the dielectric mixing model**

| Symbol | Physical property | Physical unit |
|---|---|---|
| $\varepsilon_{obs}$ | Dielectric constant (real part) measured by TDR instrument | - |
| $\varepsilon_{bound}$ | Dielectric constant (real part) of bound water at 50MHz | - |
| $\varepsilon_{free}$ | Dielectric constant (real part) of free water at 50MHz | - |
| $\varepsilon_{bound\ 1.4GHz}$ | Dielectric constant (real part) of bound water at 1.4GHz | - |
| $\varepsilon_{free\ 1.4GHz}$ | Dielectric constant (real part) of free water at 1.4GHz | - |
| $\varepsilon_{soil}$ | Dielectric constant (real part) of dry soil | - |
| $\varepsilon_{air}$ | Dielectric constant (real part) of air | - |
| $p$ | Dry porosity or saturation point | $cm^3 \cdot cm^{-3}$ |
| $w_{wp}$ | Wilting point [$cm^3 cm^{-3}$] | $cm^3 \cdot cm^{-3}$ |
| $H$ | Damping factor [-] | - |
| $w$ | Volumetric soil water | $cm^3 \cdot cm^{-3}$ |
| $v_{clay}$ | Volumetric mixing ratio of clay | $cm^3 \cdot cm^{-3}$ |
| $v_{silt}$ | Volumetric mixing ratio of silt | $cm^3 \cdot cm^{-3}$ |
| $v_{sand}$ | Volumetric mixing ratio of sand | $cm^3 \cdot cm^{-3}$ |
| $v_{SOM}$ | Volumetric mixing ratio of soil organic matter | $cm^3 \cdot cm^{-3}$ |

| OC | Organic carbon | $g \cdot kg^{-1}$ |
|----|----------------|-------------------|
| SOM | Organic matter | $kg \cdot kg^{-1}$ |
| BD | Bulk density | $g \cdot cm^{-3}$ |

The IDO model is composed of bound, mixed, and free water models, as shown in Fig. 1(a–c), respectively. The dielectric constant at the wilting point or porosity should be calculated first and then compared to the measured data in order to determine which model should be used among the Eqs.2, 3, or 4 for soil moisture estimation from the measured dielectric constant. The results of this selection for soil moisture estimation from the measured dielectric constant is displayed as shown as red dots in Fig. 1(d). The difference in the soil moisture estimation from the observed dielectric constant based on the Seyfried and IDO models is presented in Fig. 1(e). The IDO model provides larger SM values with high SOM input (purple curve) and lower SM values in low SOM input (orange curve) compared to the Seyfried model (black dotted curve). The factory setting (default probe algorithm) reflects the average SOM effect empirically in the generalized model. Even with medium-range SOM (red curve), a relatively small but more complex difference between the two approaches can be revealed in the SM estimation: lower SM estimation in wet soil and higher SM estimation in dry soil than the probe estimated (black dotted).

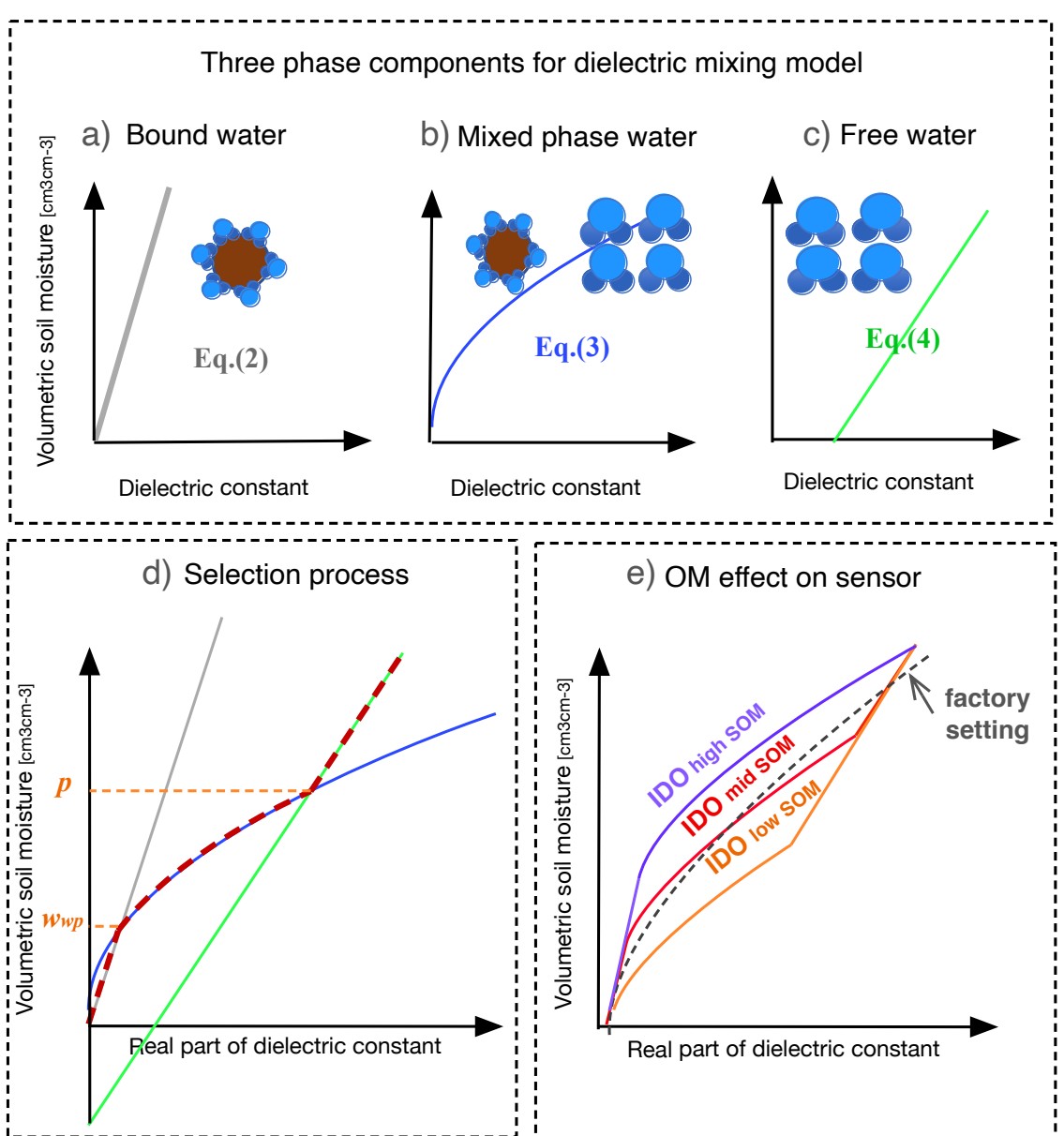

**Figure 1: Single phase relationship between (a) dielectric constant and bound water, (b) bound and free water mixture, (c) free water, (d) soil moisture estimated among those models, and (e) comparison with the polynomial-based soil probe sensor algorithm proposed by Seyfried (Seyfried and Murdock, 2004) and for organic-rich mineral soils (IDO)**

## 3. Data

First, it was necessary to determine whether including the organic matter parameter in the dielectric mixing model improves the accuracy of soil moisture estimation from the probed dielectric constant. Thus, we compared the results with the SM measured using the gravimetric method during SMAPVEX12. The SMAPVEX12 field campaign took place in 2012 (southwest of Winnipeg, Manitoba, Canada), and the SMAP SM retrieval algorithms were calibrated and validated before the launch of the SMAP satellite in 2015 (McNairn et al., 2015). During this field campaign, intensive data of the L-band brightness temperature and total radar backscatter cross-section (Tsang and Li, 1999; Entekhabi et al., 2010; Kim et al., 2014), were collected using airborne sensors. The land surface type, crop type, crop water content, soil texture (clay and sand contents) were evaluated during the experiment. Soil sampling included numerous measurements of the real part of the dielectric constant, these measurements were obtained using soil probes obtained from 16 sampling locations on 50 different agricultural fields (McNairn et al., 2015). The dielectric data was obtained approximately every 2 days between June 6 to July 17, 2012. The sampling depth of the probe is approximately 5.7cm and is representative of the soil layer relevant to the brightness temperature emission depth detectable by SMOS and SMAP (Schmugge, 1983; Jackson et al., 1997). In addition to the dielectric observations a gravimetric soil sample was obtained from each sampling field during the sampling dates. The gravimetric samples were obtained from a sampling core with dimensions of 4.7 cm diameter x 4.6 cm depth (Manns and Berg, 2014), the volumetric water content from these samples was also used for the development of calibration equations for the dielectric probes (Rowlandson et al., 2013). For comparison with our new model, we used probe measurements (real dielectric constant) as the input and volumetric soil moisture data as references (Rowlandson et al., 2013), which were simultaneously archived with microwave brightness temperature measured from airborne NASA's L-band active-passive PALS instrument. The ancillary information for this function (soil texture information) was provided by Bullock et al. (2014). At the SMAPVEX12 validation sites (Fig. 2a), the volumetric clay and sand mixing ratios for Eqs. 5, 6, 7, 11, and 12 are from the Agriculture and Agri-Food Canada (AAFC) Soil Landscapes of Canada database (Government of Canada, n.d.). The OC information was sampled from the SoilGrid250m database (Hengl et al., 2017, 2014; Poggio et al., 2021) and compared to the field estimates of the OC put forth by Manns and Berg (2014). The field samples of the OC were processed by grinding oven dried soil samples, and igniting and burning off organic mass at 375 °C. The SOM was determined from the weight difference between before and after igniting the soil samples and divided by 1.8 to convert SOM to OC (Ball, 1964; Manns and Berg, 2014; Wang et al., 2011).

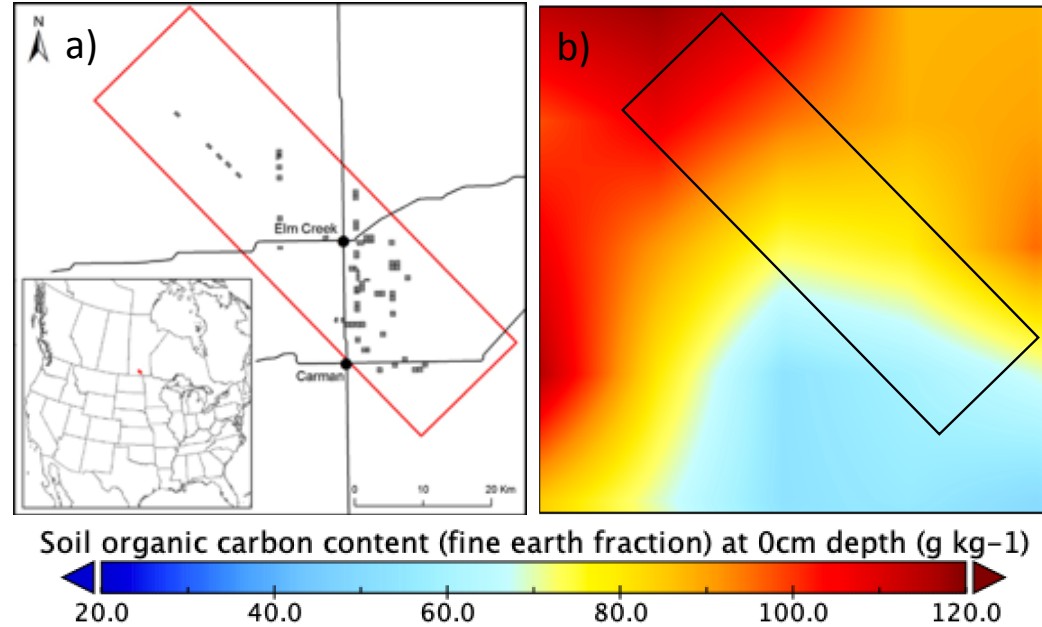

**Figure 2: (a) SMAPVEX12 validation sites (adapted from (Rowlandson et al., 2013)) and (b) calculated distribution of soil organic matter in Canada based on the SoilGrid250m database.**

There are significant range differences among the global soil organic carbon maps (Zhu et al., 2019), such as the HWSD ("Harmonized world soil database v1.2 | FAO SOILS PORTAL," 2020), SoilGrid250m (Hengl et al., 2014, 2017), WISE30sec (Batjes, 2016), and Northern Circumpolar Soil Carbon Database (NCSCD; (Hugelius et al., 2013)). Therefore, the reliability of the global soil organic maps used for local

soil moisture estimation using soil probes is still unknown. To investigate the potential limitation of global OC maps (hereafter called the $OC_{map}$ experiment), we performed a comparison of OC measurements obtained from each SMAPVEX12 site (Manns and Berg, 2014) with those retrieved from the SoilGrid205 map. As shown in Fig. 3(a), there is an offset between both datasets of $\sim 50$ g·kg$^{-1}$. The estimated OC from the map was greater and showed a wider OC range compared to the measured OC in the

SMAPVEX12 sites (Fig. 3b). This means that the SoilGrid250m (Hengl et al., 2017) estimates are, on average, more than 100 % higher than the measured data. Thus, a potential limitation of the SoilGrid250m map exists not only in the spatial pattern, but also in the overall magnitude (74.4 g·kg$^{-1}$ in average). In this study, we used OC from SoilGrid250m (without any scaling factor) for the $OC_{map}$ experiment.

We investigated the OC accuracy using one type of OC input into the new soil probe algorithm (Eqs. 2–

275 4) by performing two experiments: 1) OC entered using a SoilGrid250m map ($OC_{map}$ experiment; blue in Fig. 3) and 2) SMAPVEX12 the OC in-situ of SMAPVEX12 (red in Fig. 3).

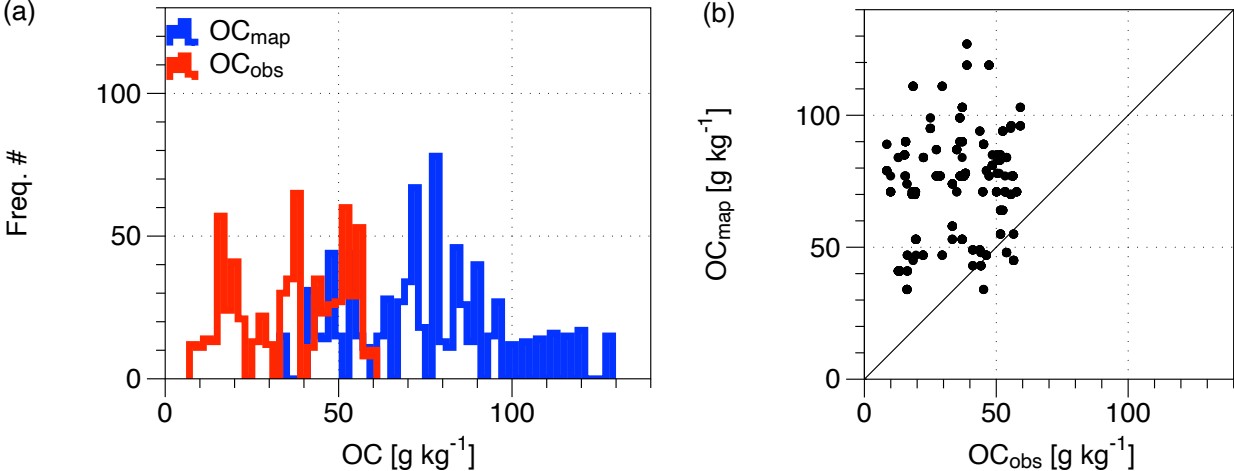

**Figure 3: Comparison between organic carbon (OC) observation from SMAPVEX12 (red) and data sampled from highly resolved SoilGrid250m map (Hengl et al., 2017) (blue) in: (a) in histogram and (b) in scatter plot.**

### 4. Calibration of portable soil-moisture sensors

The development of the calibration models is necessary for further campaigns or further extension of the global soil moisture network based on a portable soil moisture sensor. For example, calibration models (Rowlandson et al., 2013) were proposed by deriving the parameters *A*, *B*, and *C* of the quadratic function between the effective dielectric constant and soil moisture for each SMAPVEX12 station.

$$\varepsilon_{obs} = Aw^2 + Bw + C \tag{13}$$

In each site a unique set of *A*, *B* and *C* was obtained to estimate *w* (volumetric soil moisture) from the measured dielectric constant ε. It is important to verify whether these empirical models are transferable to other field sites based on physical interpretation. Therefore, we compared them with those derived from the dielectric mixing model, as shown in Table 2. The weighting function describing the attenuation of signal on probe and satellite sensor can be an exponential form basically following Beer-Lambert law where infinite attenuation of the electric field is allowed but negligible for the deeper sampling depth. On the other hand, a quadratic form can be considered as the weighting function based on the assumption of linearly decreasing refractive index scheme (Wilheit, 1978) so that the emission can be assumed to be zero from the deeper sampling depth. In this study, as shown in Table 2, we assumed the Beer-Lambert law to consider the attenuation effect by applying the damping factor 0.8 applicable both for probe and satellite remote sensing. More detailed derivation associated with the damping factor can be found in the previous study (Park et al., 2017).

**Table 2. A, B, and C parameters of the relationship between the effective dielectric constant and soil moisture adapted from (Park et al., 2017) with damping factor $H$ (0.8) ; dielectric constant for free ($\varepsilon$free), bound water ($\varepsilon$bound) and soil mineral including organic matter ($\varepsilon$soil).**

| $w$ range | A | B | C |
|---|---|---|---|
| $w < w_{\text{wp}}$ | 0 | $(\varepsilon_{bound} - \varepsilon_{air})H$ | $\big((1-p)\varepsilon_{soil} + p\varepsilon_{air} - 1\big)H + 1$ |
| $w_{\text{wp}} < w < p$ | $\dfrac{\varepsilon_{\text{free}} - \varepsilon_{\text{bound}}}{p - w_{\text{wp}}}H$ | $\left(\dfrac{p\varepsilon_{\text{bound}} - w_{\text{wp}}\varepsilon_{\text{free}}}{p - w_{\text{wp}}} - \varepsilon_{air}\right)H$ | $\big((1-p)\varepsilon_{soil} + p\varepsilon_{air} - 1\big)H + 1$ |
| $p < w$ | 0 | $(\varepsilon_{\text{free}} - \varepsilon_{\text{soil}}) \cdot H$ | $\varepsilon_{\text{soil}}H - H + 1$ |

We observed that when the wilting point and porosity increased with increasing OC [according to Eqs. (11–12)], $A$ and $B$ increased and decreased, respectively, as shown in Fig. 4. The results of this matching (Fig. 4) showed that $A$ and $B$ used in the quadratic function computed for SMAPVEX12 can be parameterized with soil texture, wilting point, porosity, and the bound and free water dielectric constants. Additionally, the $C$ parameter indicates the effective dielectric constant of the mixture of dry organic
matter (approximately 1.2 (Savin et al., 2020)) and solid mineral soil (3-5); ideally, the C parameter value should decrease with an increase in OC. Notably, the clay content was also positively correlated with an increase in OC in the SMAPVEX12. Therefore, owing to the simultaneous increase in clay content, which is characterized by a high dielectric constant, the sensitivity of the $C$ parameter to OC variation (decreasing pattern in $C$) is nullified, as shown in Fig. 4(c). Furthermore, because $C$ perfectly represents
the dielectric constant of dry soil, it should be greater than 1, which is the real part of the dielectric constant of a vacuum. Based on this physical constraint, the previous $C$ (gray points in Fig. 4) is unrealistically low (less than that of the vacuum state) in the higher SOM range. The minimum C is $(1-p)\varepsilon_{\text{soil}}$ among three $w$ ranges [Eq. (4)] because the following order is always true [$(1-p)\varepsilon_{\text{soil}} < (1-p)\,\varepsilon_{\text{soil}} + p < \varepsilon_{\text{soil}}$] and it is larger than 2 as shown in Fig. 4 C. This shows that the proposed IDO computes a more realistic value
of dielectric constant for organic-rich mineral soil.

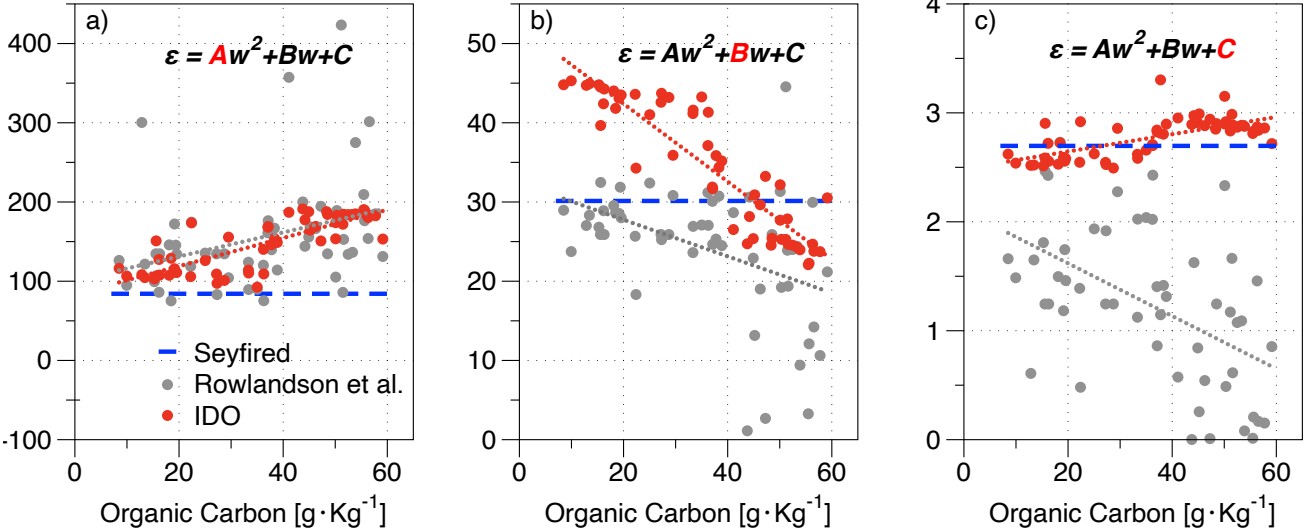

**Figure 4: Relationship between soil organic carbon measurements (x-axis) and calibration parameters (*A*, *B* and *C*) (y-axis) relating between measured dielectric constant (*ε*) and volumetric soil moisture (*w*): (blue dash lines) *A*, *B*, *C* which are not sensitive to OC measurements (Seyfried approach); (gray dots) *A*, *B*, *C* which are empirically obtained (Rowlandson et al., 2013); (red dots) *A*, *B*, *C* which are physically simulated by proposed IDO which applies the wilting point and porosity as functions of sand and clay volumetric mixing ratios as well as soil organic carbon with the damping factor applied.**

### 5. Results

This study aimed to mitigate a significant discrepancy found between volumetric soil moisture estimated by soil probe sensor (considered as ground truth for the validation of land surface modelling and remote sensing) and the gravimetric soil moisture. Therefore, in this section, the new approach proposed in the section 2 investigated whether the accuracy of the new sensor algorithm can be improved compared to the existing probe algorithm. Firstly, looking at the Fig. 5(a) the current issue in the probe SM estimates was well displayed in terms of the matching pattern of the gravimetric soil moisture with the measured dielectric constant. It showed that the existing probe soil moisture (red dots in Fig.5(a)) couldn't follow both features that appeared in the measurements (the significant scattering degree and the distinct varying patterns in dry and wet condition). This is a fundamental limitation of the traditional polynomial function, the Seyfried model as well as a two-mode system (mineral or organic (peat) soil), as proposed by Topp et al. (1980) or Roth et al., (1992).

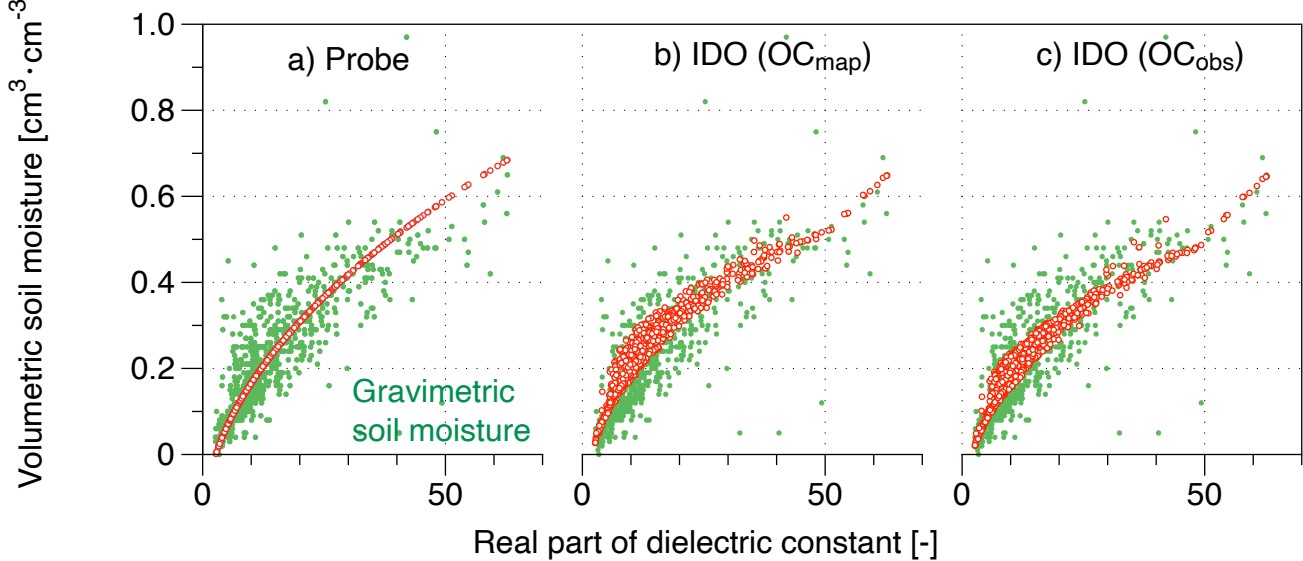

**Figure 5: Scatter plot between probe measurements of the real part of the dielectric constant (x-axis) and volumetric soil moisture (y-axis) measured by gravimetric method (green dots in a, b, c), Seyfried model (red dots in a), IDO with SOM taken from SoilGrid250m (Hengl et al., 2017) (red dots in b) and IDO using OC measured during SMAPVEX12 (Manns and Berg, 2014) (red dots in c).**

On the other hand, the IDO, with soil organic carbon considered, allowed us to compute SM with a similar scattering pattern comparable to that measured by gravimetric method. It means that soil organic carbon is a critical factor for the application of the soil moisture sensors from portable to satellite based. In regards to the shape appearing in the scattering pattern, IDO captured the distinctively curved edge in the low and high-end points close to the values of 12 and 50, respectively, in the x-axis for the real part of the dielectric constant. The only difference between b) and c) at Fig.5 is OC input, originated from SoilGrid250m or from in-situ obtained during SMAPVEX12, respectively, with the same input of clay and sand mixing ratio from SMAPVEX 12. This pattern is probably related to the transition moments from bound to mixed (a to b in Fig. 1) and from mixed to free water states (b to c in Fig.1), which is very interesting evidence indicating that soil probes can detect critical soil parameters such as wilting point and soil porosity based on the accumulated dielectric measurements of certain sites.

Even though the shape of SM scattering estimated from the measured dielectric data (x-axis) became similar to the one appearing in the gravimetric soil moisture, it is also required to investigate whether the actual improvement in the SM accuracy has been achieved via the point-by-point comparison with the gravimetric data. This analysis was illustrated in the Q-Q plot in Fig.6. It displayed that the scattered uncertainty shown in Fig. 6(a) of the current soil probe algorithm can be reduced by IDO approach as (b) and (c). The scatter error shown in Fig.6(a) slightly converged into a 1:1 line when the IDO adapted the OC map as input (b) and further improved with a narrower scattered error pattern with OC in-situ (c). This result further supported that the OC variability with the proposed model can mitigate the uncertainty in SM estimation of the current dielectric-based soil moisture sensor network.

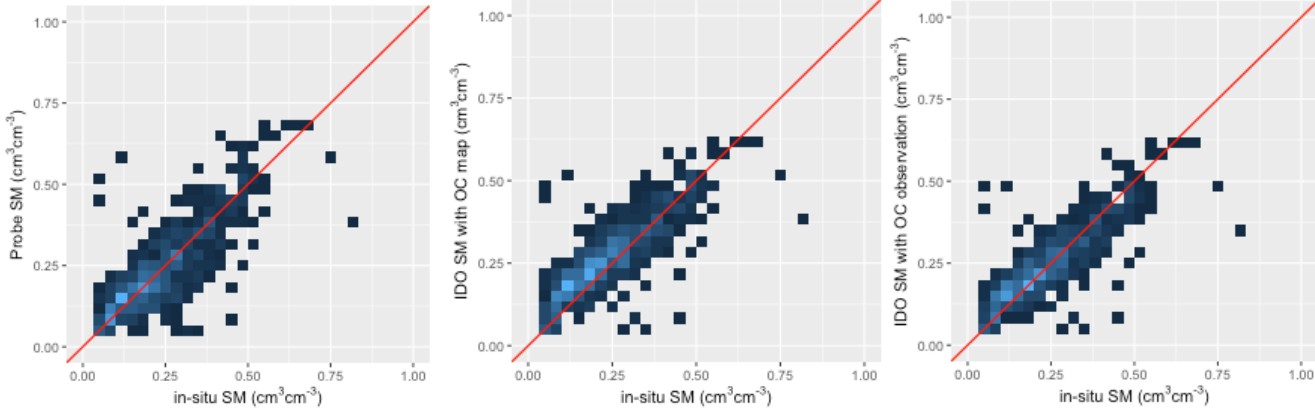

**Figure 6: Performance of soil moisture probe algorithms in terms of scattering degree to the gravimetric measurements (x-axis); soil moisture estimates (y-axis) using a) 3rd order polynomial approach (Seyfried and Murdock, 2004; Bell et al., 2013), b) the proposed inverse dielectric mixing model (IDO) with the variational soil organic matter (SOM) sampled from the SoilGrid250m map (Hengl et al., 2017) and c) the same algorithm but with SOM measured from SMAPVEX12 (Manns and Berg, 2014).**

In Fig. 7, we investigated more characteristics of SM uncertainty; how the biases of SM estimated by the conventional probe algorithm are related to the in-situ OC and whether they can be mitigated by the proposed algorithm with the OC measurements. Fig. 7(a) shows that both negative and positive biases are affected by the IDO. Fig. 7(b), obtained by spreading out the histogram according to the degree of SOM, provides an in-depth analysis on how these biases are distributed according to the measured SOM. This shows that the negative bias in the high SOM range was reduced because the polynomial function of the conventional probe algorithm presented in Fig. 1(e) tends to overestimate the SM in the cases of lower SOM and underestimate the SM in cases of higher SOM (as compared to the proposed multiphase model).

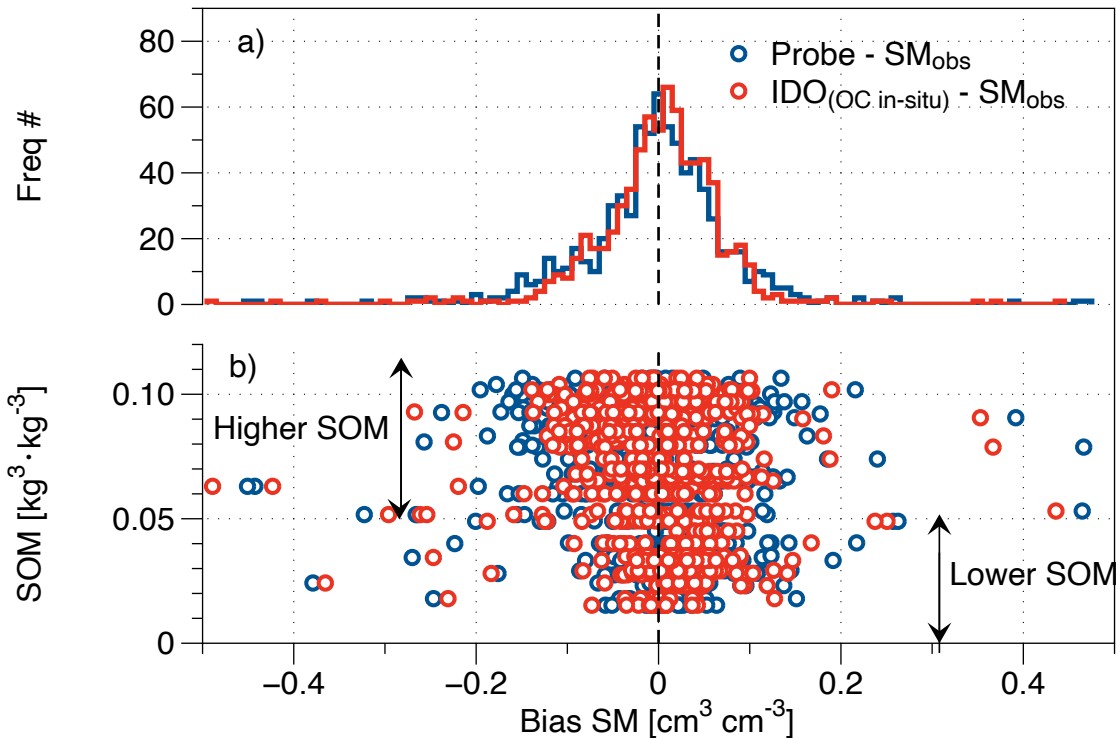

**Figure 7: (a) Histogram of soil moisture (SM) bias and (b) its scatter relationship according to soil organic matter (SOM) converted from in-situ organic carbon (OC)**

The importance of accurate and highly resolved organic carbon data in soil moisture estimation from portable soil sensors is highly evident from the statistical validation presented in Table 3. The results confirmed that the IDO performs better than the traditional probe algorithm based on a 3$^{rd}$ order polynomial function especially with the OC measured in the SMAPVEX12 field campaign (with a maintained spatial variability); RMSE = 0.0727 cm$^3$·cm$^{-3}$, correlation of 0.848, and bias of 0.0001 cm$^3$·cm$^{-3}$.

**Table 3. Validation of soil moisture obtained from Probe (Probe SM), organic-rich mineral soil based on SoilGrid250m organic carbon map (IDOmap), and SMAPVEX12 OC in-situ observation (IDO$_{obs}$)**

|  | Bias | RMSE | Correlation |
| --- | --- | --- | --- |
| Probe | -0.0042 | 0.0824 | 0.824 |
| Proposed algorithm with organic carbon map (IDO$_{map}$) | 0.0222 | 0.0789 | 0.835 |
| Proposed algorithm with in-situ organic carbon (IDO$_{obs}$) | 0.0001 | 0.0727 | 0.848 |

The results in this section demonstrated that the wilting point and porosity which emerged in paring the gravimetric soil moisture and the dielectric measurements, could be detected also by the new model. Also, it is proven that the volumetric soil moisture could be estimated from the sensor more accurately in terms of bias, RMSE and correlation analysis. It means that our approach can provide more accurate soil moisture probe algorithm than currently used in various soil moisture networks such as USCRN (US Surface Climate Observing Reference Networks) and the SMAPVEX field campaigns. In the boreal forest and Alaska Tundra region with abundant SOM, our study can deliver a significant effect to the validation and conclusion of the previous studies in land surface modelling and microwave satellite remote sensing, which used the probe soil moisture as a reference data.

## 6. Summary and discussion

In this study, we proposed an inverse dielectric mixing model for a 50-MHz soil sensor for agricultural organic-rich mineral soil. The 50MHz sensor is a prevalent frequency band for soil moisture probes. (Cosh et al., 2021) found that in North America soil sensors using this waveband occupied 40% of the soil moisture networks (10 of 25 including USCRN) and 53% of sensors (1021 of 1923 locations). Therefore, the proposed algorithm has potential to contribute significantly to the accuracy of the soil moisture estimates derived from current in-situ soil moisture measurements. Furthermore, since the SMAPVEX also used 50 MHz sensors, it is anticipated that the accuracy of the calibration and validation of the SMAP related soil moisture algorithms will be increased. The proposed model is composed of three nonlinear functions that are mathematically capable of describing the physical behaviour, including the effect of the organic matter content. In this model, we proposed a physical mixing approach of organic matter in dry soil and improved the wilting point and saturation point. This derivation also can be applied to other bands for Capacitance sensors (5TE (70MHz), Wet (20MHz), Time-Domain-Reflectometry (TDR) (TDR100/200 (1450MHz), SoilVUE-10 (1450MHz) and satellite sensors SMAP (L-band) and SMOS (L-band) (AMSR-E (JAXA) X/C, Sentinel-1 (ESA) (C)). It is also noticed that the applied organic matter carbon data sampled from SMAPVEX12 sites (36 g·kg$^{-1}$) was half that of the OC map (74 g·kg$^{-1}$). The validation results demonstrated a higher performance of the new model. Regardless of the small amount of OC, its effect improved the performance of the SM estimation, which was demonstrated via the IDO proposed in this study. We compared the obtained soil-moisture retrievals with improved RMSE (13% ↓), slightly stronger correlation (3%↑), and lower bias (90%↓) using the new model and gravimetric soil moisture data. But still the coverage of the simulated pattern over the measured points was smaller. Therefore, we sought out a potential further improvement based on the additional experiment designed with SOM varying within the proposed model. The simulation based on the conventional polynomial function (red curve in the Fig. 8 (a)) could not reduce the innate uncertainties and the IDO proposed in this study could resolve this issue. However, the red dots simulated with IDO (Fig. 8(b)) covered over the measured green dots insufficiently. Therefore, in order to activate this weak pattern, we performed the experiments to impose a more dynamic OC estimate to investigate whether greater or less SOM can cover a similar boundary of the measured distribution through the IDO model. The results showed that the piecewise pattern of SM simulated with the proposed approach well covered the measured pattern with imposing lower (1 %) to higher (30 %) SOM.

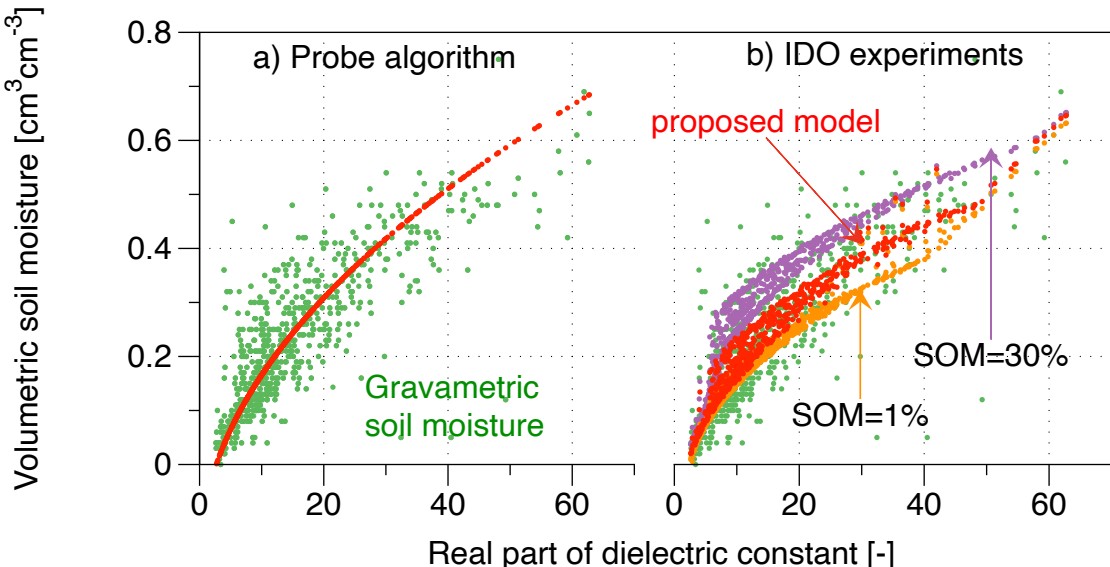

**Figure 8: Investigation of the similarity of the scatter pattern between the measured dielectric constant and the soil moisture: (a) obtained from the gravimetric measurements and (b) experimentally simulated with extreme soil organic matter (SOM) from 0 % to 30 %.**

Because the SOM is translated from OC with a conversion factor (1.8) in this study, the improvement might have been not sufficient. A realistic estimation of the conversion factor ($f_{oc}$) in Eq. (10) varying from 1.25 up to 2.5 might be a possible solution for this. In addition, the IDO is a model able to replace the calibration factors $A$, $B$, and $C$ of Eq. (13) with the soil properties presented in Table.2. Overall, the proposed more physics based IDO can replace the current soil probe sensor algorithm, which does not incorporate the importance of organic matter variability.

A significant improvement could not be shown probably due to two reasons: the instrumental error in measuring OC from soil sample or the constant OC to SOM conversion factor (1.8 for all soil samples). In addition, uncertainty can be suspected from other sources, such as clay or sand contents or soil salinity (assumed to be 0 % in this study) used in the IDO. These effects on the dielectric measurements and their uncertainties probably served as the limitation of further improvement by the IDO. Therefore, for the potential users to apply our approach, note the following range of SOM applied: our study was validated in 1% - 15 % and performed the sensitivity experiment in 1% - 30% SOM.

Nevertheless, the results regarding the adaptation of in-situ OC in our study demonstrated that the accuracy of the SOM input for IDO is critical for the accuracy of SM estimation from the probe sensor. In previous studies (Topp et al., 1980; Roth et al., 1992; Bircher et al., 2016), in the organic mode or the peat soil, the dielectric constant and soil moisture relationship is calibrated to be able to simulate the dielectric constant lower than mineral soil with given soil moisture. These results are consistent with our study, which showed decreasing dielectric constant value in higher SOM by increasing bound water fraction due to higher wilting point (wp). Therefore, if we have more information about the dielectric constant of perfectly dried peat soil and a more accurate model for the wilting point and porosity of this soil, our model will be able to cover soils from mineral to peat regions to obtain more accurate global soil

moisture. In addition, if we improve this model toward a frequency dependent model in the future study, the existing and future probe measurements obtained in various frequencies will be able to contribute more extensively for the calibration and validation of satellite and model.

Author contributions:

CHP developed the algorithm. CHP, AB, MHC, AC and HM designed the study. HM collected the soil organic carbon in-situ data. RZ provided the mathematical corrections. CHP conducted the validation and the analysis. JH, AB, JL and VW provided guidance on the research direction. CHP wrote the manuscript. All authors reviewed the manuscript.

Acknowledgement:

We acknowledge anonymous reviewer and the editor who provided valuable feedbacks to improve this work. This study was funded by the Korea Meteorological Administration Research and Development Program "Development of Climate Prediction System" under Grant (KMA2018-00322). The USDA is an equal opportunity employer and provider. A partial contribution to this work was made at the Jet

Propulsion Laboratory, California Institute of Technology under a contract with National Aeronautics and Space Administration and a National Research Foundation of Korea Grant from the Korean Government (MSIT) (NRF-2018R1A5A1024958).

Funding:

Declaration of Interest:  The authors declare that they have no conflict of interest.

**Appendix A**

Based on the computation of the bulk density for organic soils, the volumetric mixing ratio of soil organic matter can be derived as shown in Eqs. (A.1)-(A.7).

$$BD_{soil} = \frac{M_{MI}+M_{SOM}}{V_{MI}+V_{SOM}} = \frac{BD_{MI}\times BD_{SOM}}{(1-SOM)\times BD_{SOM}+SOM\times BD_{MI}} \tag{A.1}$$

$$\frac{V_{SOM}}{V_{MI}+V_{SOM}}\frac{M_{MI}+M_{SOM}}{V_{SOM}} = \frac{BD_{MI}\times BD_{SOM}}{(1-SOM)\times BD_{SOM}+SOM\times BD_{MI}} \tag{A.2}$$

$$\upsilon_{SOM} = \frac{V_{SOM}}{M_{MI}+M_{SOM}}\times\frac{BD_{MI}\times BD_{SOM}}{(1-SOM)\times BD_{SOM}+SOM\times BD_{MI}} \tag{A.3}$$

$$\upsilon_{SOM} = \frac{M_{SOM}}{M_{MI}+M_{SOM}}\frac{V_{SOM}}{M_{SOM}}\times\frac{BD_{MI}\times BD_{SOM}}{(1-SOM)\times BD_{SOM}+SOM\times BD_{MI}} \tag{A.4}$$

$$\upsilon_{SOM} = \frac{SOM}{BD_{SOM}}\times\frac{BD_{MI}\times BD_{SOM}}{(1-SOM)\times BD_{SOM}+SOM\times BD_{MI}} \tag{A.5}$$

$$\upsilon_{SOM} = \frac{BD_{MI}}{\frac{(1-SOM)}{SOM}\times BD_{SOM}+BD_{MI}} \tag{A.6}$$

$$\upsilon_{SOM} = \frac{1}{\left(\frac{1}{SOM}-1\right)\times\frac{BD_{SOM}}{BD_{MI}}+1} \tag{A.7}$$

where, $M_{MI}$ [kg] and $M_{SOM}$ [kg] are mass of mineral and soil organic matter; $V_{MI}$, [cm$^3$] and $V_{SOM}$ [cm$^3$]

are volume of mineral soil and soil organic matter; $\upsilon_{SOM}$ [cm$^3$ ·cm$^{-3}$] and SOM [kg·kg$^{-1}$] are volume and

mass mixing ratio; and $BD_{MI}$ [kg·cm⁻³] and $BD_{SOM}$ [kg·cm⁻³] are bulk density of mineral and organic matters, respectively.

## Appendix B

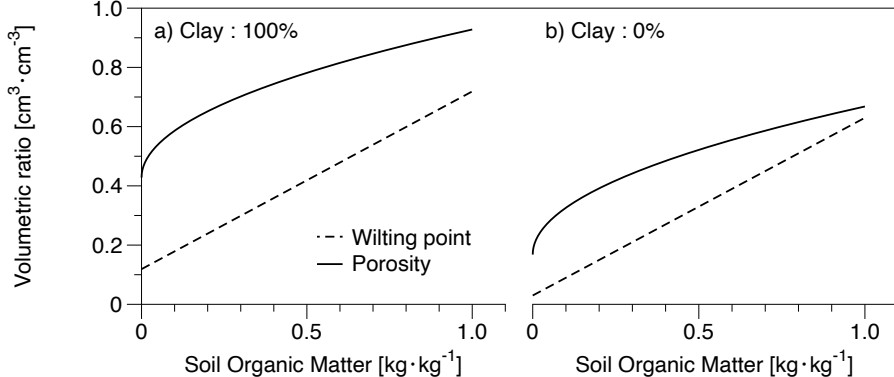

**Figure B1 Simulations of wilting point, Eq.(11) improved of Park et al., 2019 and porosity, Eq.(12), proposed in this study which are function of soil organic matter at extreme case a) volumetric clay mixing ratio 100% and b) 0% of the total mineral within soil**

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
