# Peer review of "An inverse dielectric mixing model at 50MHz that considers soil organic carbon"

_Hydrology and Earth System Sciences, 2021_

## Author Response (AR1)

I sincerely appreciate the thoughtful and productive comments of the anonymous reviewer and the editor, Dr. Gerrit de Rooij. Based on the reviewer's opinions we corrected the mathematical errors in the equations. Furthermore, we clarified more effectively the benefit of our study in the development of more accurate soil probe and satellite remote sensing algorithms. We responded in detail as the following. In the revised manuscript, the reviewer's and the editor's points were highlighted with ==yellow== and ==cyan==, respectively.

**Review 1**

This study developed a general inverse dielectric mixing model that can be applied to retrieve soil moisture from in-situ dielectric data. Considering organic carbon is the novel part of this approach. Overall, I think it is an interesting study. A reliable dielectric mixing model for organic soils is highly needed in order to get accurate soil moisture estimates in the high latitudes from both in-situ sensors or spaceborne microwave sensors. However, there are quite a number of places in the mathematical expressions that should bee carefully examined.

The data used for validation also have a very limited range of organic carbon content (OC<0.06 g/g), which are generally not classifed as organic soils (rather they are organic-rich mineral soils). If the authors claimed this approach is going to help the soil moisture retrieval in the boreal and Arctic region, more data with a wider range of OC content are needed. Comparison with previous approaches that incorporated SOC (such as Bircher et al. 2016) should be also included and discussed.

 Bircher, S., Andreasen, M., Vuollet, J., Vehviläinen, J., Rautiainen, K., Jonard, F., Weihermüller, L., Zakharova, E., Wigneron, J.-P., and Kerr, Y. H.: Soil moisture sensor calibration for organic soil surface layers, Geosci. Instrum. Method. Data Syst., 5, 109–125, https://doi.org/10.5194/gi-5-109-2016, 2016.

➔ We thank the reviewer for their thorough assessment and helpful comments. We responded to the two main questions above in the reviewer's specific comments.

Specific                                                                                          comments:

1. Comments on the euqations:

(1) Eq (5) and (6):  it is confusing whether this applies to 50MHz, or any frequency ranging from 1.4GHz to 50MHz. Also please provide the original references for these two equations.

➔ Thank you for the comment. Our application is targeting for 50MHz sensor. To avoid the confusion, we corrected the paragraph (p.4, lines 135-141). Also, we added the relevant references for Eq.(5) and (6).

(2) Eq. (8) does not seem correct to me. The correct equation should be: v_som = 1/((1/SOM-1)(BD_SOM/BD_soil)+1), where in the original equation, the BD_soil and BD_som should switch. The authors should also clearly define "BD_soil" and "BD_SOM". If "BD_soil" is the soil bulk density, it varies greatly based on the SOM concentration, and using a constant value is not proper. Rather, I think here "BD_soil" and "BD_SOM" should mean the specific density of "mineral" and "organic matter" part of the soil solids. Please double check this.

➔ Thank you very much for pointing out this error. The mineral and organic matter should be switched as shown in the corrected Eq.(8) (p.5, lines 152-156). Furthermore, we specified the BD_SOM and BD_MI are constant based on the Eq.(9) with assumption of 0 and 56% OC cases (p.5, lines 177-180) which are called as "pure mineral" and "pure organic matter". We also added the detailed derivation of Eq.(8) in the appendix A (p.18-19, lines 480-500).

(3) Eq. (9): if this equation is used to estimate the soil bulk density for organic soils, it is not applicable to highly organi soils. E.g. when SOM=1 (or OC=~0.58), BD_SOM>2 g/cm3. The original reference does not include data with OC > 20%. This is also a common problem in the empirial equations derived for organic soils or organic-rich soils.

➔ The bulk density function is used for the calculation of "pure" organic matter and "pure" mineral particles. As you mentioned for very high OC, the bulk density becomes unrealistically large. Therefore, we replaced it to more proper and well-known equation to estimate their pure density proposed by Hossain et al., 2015 as Eq.(10) (p.5, line 162) and further explained how to calculate $BD_{MI}$ and $BD_{SOM}$ using this equation (p.5, line 177-180).

(4) There is no equation (10).

➔ Thank you for the comment. We re-ordered the equation numbers properly.

(5) Eq. (11) & Eq (12): similar as above, the authors should mention under what OC range those euqations can be applied to, esp. the estimate of wilting point. For soils with high SOM concontration, Eq. (11) gives a very large estimate, which is even close to the porosity provided by Eq. (12).

➔ In Appendix B (p.19, lines 502-507), we avoided the issue that the wilting point becomes higher than porosity in high SOM by improving Eq.(11) and (12). In the new parameterization, we could have porosity which is larger than wilting point in all SOM and clay ranges.

2. Line 185: the authors should provide a brief description how the field estimates of OC was derived.

➔ We added the description about OC sampling (p. 8, lines 237-241).

3. SoilGrids data: Does the author use SoilGrids 1km or SoilGrids 250m? The authors indicates SoilGrids1km in one place, while it says SoilGrids250m data were used in another place. These two datasets can be quite different in terms of OC estimates. Bedies, SoilGrids1km provide OC estimates at certain depths, while SoilGrids250m provides OC estimates for different soil intervals. Please clarify. Fig.2 (b): please provide colorbar for the OC map.

➔ Thank you for pointing out this error. Because we used SoilGrid250m in our study, we fixed the SoilGrids1km to SoilGrids250m (p.9, line 249). We added the color bar properly in Fig.2 (p.9, line 244).

4. Comments on the results:

  (1) Fig. 5 (a) what are the results derived using the Seyfried model? The red dots?

➔ Yes, the Seyfried model is indicated with the red dots. We corrected the black dots to the red dots in Fig. 5 (a) (p. 13, line 331).

  (2) Fig.6 & 7: the reduction of the uncertainty in SM estimates is relatively limited. It may be partly due to a narrow range and also a low amount of SOM in the soil samples in the study area. Therefore, it needs additional investigation whether this method applies to highly organic soil (e.g. SOM>30%), prevalent in the boreal and Arctic region, and how it performs if it is applicable.

➔ I agreed with you about your comments that in the low or medium OC range, the previous study (Bircher) was hard to demonstrate the improvement by OC consideration and we made the significant difference as you mentioned in the last comment. Therefore, I believe the improvement is meaningful without further experiments.

Furthermore, the direct comparison with the Bircher's study is difficult for this study because of two different conditions; 1) the OC measurements (the OC measurements of Bircher's study is obtained from soils highly covered with moss, which show very different characteristics from the one obtained from our SMAPVEX12 agricultural soil and 2) the dielectric measurements (Bircher's study used the theta probe which the frequency used in is 100MHz).

Therefore, we decided to investigate the possibility of further improvement for higher OC regions in the future study.

I would think this method is more general and has a high potential applicable to those conditions. However, the parameterization (including the witling point, porosity) needs additional improvement.

➔ After improving the wp and p function based on your previous comment (p.5-6, lines 186-188), we could update the validation score properly in the table 3 (p.15, line 392).

It will be also helpful if the authors can compare their results with the previous methods that particularly incorporates organic carbon content. For example, in Bircher et al. 2016, the data do not show substantially dielectric differences in soils with SOM<30%, while this study shows even a small amount of SOM (SOM<~10-11%) can make a significant difference in the relationship between SM and dielectric constant.

➔ As the reviewer's suggestion, the comparison to the study (Bircher et al. 2016) will be very helpful to demonstrate the benefit of our approach for 50MHz soil moisture sensor. However, as mentioned in above response, in the current study it is not directly applicable due to the difference in the type of organic soil and in the frequency of the soil probe sensor. I agreed with you about your comments that in the low or medium OC range, the previous study (Bircher) was hard to demonstrate the improvement by OC consideration and in our study, we made the significant difference as you mentioned in the reviewer's last comment. Therefore, I believe the improvement is meaningful without further experiments. We decided to investigate the possibility of further improvement for higher OC regions in the future study.

**Editor (Dr. Gerrit de Rooij)**

The reviewer noted various occurrences of lax editing that can be easily fixed. It is the responsibility of the authors, not of the reviewers, to eliminate such errors (preferably before submitting the paper). I therefore ask you to carefully go through your paper again and also remove errors that the reviewer did not highlight.

➔ Including the errors pointed out by the reviewer, we found the additional mathematical errors in some formulas specially by our newly joined co-author. In this revision have corrected all of them properly (p.3, lines 12 and 13; p.4, lines 121-126; p.5, line 160; p.11, line 292).

The reviewer is concerned about the limited range of organic matter content in the soils you consider and objects to the term 'organic soil', which you use on multiple locations in the paper, including the abstract. I agree with the reviewer that the term 'organic soils' is usually reserved for peaty soils with limited mineral content. The reviewer also points out your results disagree with results reported in the literature for similar, relatively low, organic carbon contents. From you reply it appears you have confidence in your data and you are not prepared to do more experiments on

soils with a higher organic carbon content. If I understand your reply correctly your position is that the focus on organic-rich mineral soils (e.g., not 'organic soil' per se) is important enough in and of itself to warrant the dedication of the paper to them. If that is the case, your paper would improve if you dedicate a bit more text to clarifying the contribution and relevance of your work, and avoid using the term 'organic soil', because this can create misunderstandings among readers.

➜ Thank you very much for your comments. We agreed with the reviewer's and the editor's opinion. The 'organic soil' used in our manuscript is not proper expression and might cause the confusion to readers. Therefore, according to the editor's comment, we replace the term 'organic soil' into 'organic-rich mineral soil' which is highlighted with cyan color in p.1, line 18; p.3, line 82; p.4, line 144; p.5, line 176; p.6, line 190; p8, line 213; p.11, line 308; p.12, line 330; p.15, line 389; p.18, line 481; p.19, line 507.

Your reply already points to the relevance of your work to some degree, so you appear to be developing a suitable line of argumentation that you can include in the revised text.

➜ We emphasized the importance of our study relating to the current soil moisture probe network in the discussion. For example, the improvement by our study can be anticipated on over 50% locations of the current sensors installed in the United States (page 16, lines 404-410). Furthermore, we discussed that the proposed mixing approach for dry organic soil is applicable to the other soil probe sensors in the different range of frequencies (p.16, lines 404-410).

It will also help if you can expand the discussion of your results in view of earlier findings reported by others.

➜ In the discussion, we investigated about whether the results presented in our study is consistent to the previous studies (Topp et al., 1980; Roth et al., 1992; Bircher et al., 2016) (p.17-18, lines 451-460). As a result, our estimation of soil moisture from the dielectric constant showed the similar pattern with the one estimated by previous results, and it could be explained by the bound water fraction of organic soil in our study.

---

## Referee Report (RR1)

Overall comments:

I think the authors addressed my comments, and the manuscript was improved to some extent. I agreed it is important to have a more general model that can be applied to organic-rich mineral soil globally. In the new version, the authors do emphasize that the proposed model and its validation are mostly done for soils with low range of SOM values. I would think that it remains uncertain whether this model applies to soils with very high SOM content, particularly the validity of the wilting point estimation.

1. please update the statistics in the abstract to be consistent with the results

2. Introduction, P 2, Line 75: Are the authors trying to say that we need dynamic SOC maps, rather than statical maps available from current datasets? I agree that it is important to get global SOC estimates from satellite missions, especially in data-sparse regions where the currently statistical-based SOC maps may fail. Yet I am not sure whether SMAP data can be very helpful in this regard, esp considering its coarse resolution.

3. Eq. (11-12): wilting point is a critical parameter of the proposed dielectric model, which determines the ratio of boundwater to free water.

   Are there any references to back up the new equation (i.e. Eq. 12)? Or the authors simply adjusted the original equation to avoid wilting point larger than porosity? Some references supported a rather low witling point of highly organic soils (e.g. Verry et al 2011, Physcial Properties of Organic Soils), so I remain doubtful about the validity of the new equation for peaty soils. I suggest the authors noted in the ms on the range of SOM levels that the equations may apply to.

4. Please note that a new SoilGrids250m dataset (v2.0) is now available (Poggio et al., Soils, 2021).

5. Appendix A: please relable the equation as A.X to avoid confusion. Also, some expressions are awkward. E.g. P8, Line 222: what does "back-scattering albedo" mean? There are still errors in the format (e.g. P8, Line 225). Please go through the paper carefully.

---

## Author Response (AR2)

We thank the reviewer for providing valuable comment to improve our manuscript. Our response can be found in gray paragraphs.

Overall comments:

I think the authors addressed my comments, and the manuscript was improved to some extent. I agreed it is important to have a more general model that can be applied to organic-rich mineral soil globally. In the new version, the authors do emphasize that the proposed model and its validation are mostly done for soils with low range of SOM values. I would think that it remains uncertain whether this model applies to soils with very high SOM content, particularly the validity of the wilting point estimation.

We also agreed that general model which can cover from mineral to peat soil is important to be developed. Unfortunately, in this study we could not present the results for the high OC cases because we focused on the development of the soil probe sensor for the 50MHz frequency and there are no dielectric measurements in this wavelength obtained from peat soil in the previous literature including Bircher's study.
As stated in the discussion section, we will improve this model as a frequency dependent model to compare the simulated with the measured dielectric constants obtained from very high organic or peat soils such as Bircher's study. However, the main idea of the current study will be consistently applied in the future study that increases OC, increases bound water fraction, and consequently decreases dielectric constant regardless of the frequency difference of the soil probe.

1. please update the statistics in the abstract to be consistent with the results

Thank you very much for pointing out this error. We updated the validation statistics (p. 1, lines 28-29).

2. Introduction, P 2, Line 75: Are the authors trying to say that we need dynamic SOC maps, rather than statical maps available from current datasets? I agree that it is important to get global SOC estimates from satellite missions, especially in data-sparse regions where the currently statistical-based SOC maps may fail. Yet I am not sure whether SMAP data can be very helpful in this regard, esp considering its coarse resolution.

We believe that SMAP might play very important role in obtaining the spatially and temporally varying SOC map which has not been obtained before from satellite measurements. The reason for this belief is that regardless of the coarse resolution of SMAP measurements it can provide an unprecedented and unique benefit to estimate SOC for following three reasons.
1) Microwave can only detect SOC underneath vegetation which other shorter wave sensors cannot perceive.
2) The temporal dynamic of OC evolution obtained even from low-resolution satellite image will be helpful in various modeling and observation studies.
3) The limitation of the low-resolution issue can be overcome by recent downscaling approaches, machine learning methods to make a synergy with other ground, spaceborne and satellite data.
We added above description in P. 2-3, lines 76-84.

3. Eq. (11-12): wilting point is a critical parameter of the proposed dielectric model, which determines the ratio of boundwater to free water.
Are there any references to back up the new equation (i.e. Eq. 12)? Or the authors simply adjusted the original equation to avoid wilting point larger than porosity? Some references supported a rather low witling point of highly organic soils (e.g. Verry et al 2011, Physical Properties of Organic Soils), so I remain doubtful about the validity of the new equation for peaty soils. I suggest the authors noted in the ms on the range of SOM levels that the equations may apply to.

There are various backup studies (Saxton, Chadburn, Schaap) to show that greater SOM increases the wilting point and porosity as our proposed equations. This relationship between the organic matter and the soil parameters might become more complex in organic rich soil if the type of OC is also important and the dominancy for the OC type change in high SOC region. In this regard, this complex relationship considering sapric, hemic and fibric as referred by the reviewer ((e.g. Verry et al 2011, Physical Properties of Organic Soils)) should be defined in the future. We added this explanation in p. 6, lines 196-202.

As the reviewer's comments, we added the range of OC for the validation (1% - 15 %) and the sensitive experiment (1% - 30% SOM) for the clarification of the future users for our approach in p.18, lines 458-460.

4. Please note that a new SoilGrids250m dataset (v2.0) is now available (Poggio et al., Soils, 2021).

Thank you very much for the relevant reference. We added this in p. 9, line 251.

5. Appendix A: please relable the equation as A.X to avoid confusion. Also, some expressions are awkward. E.g. P8, Line 222: what does "back-scattering albedo" mean? There are still errors in the format (e.g. P8, Line 225). Please go through the paper carefully.

We added more detail description for "back-scattering albedo" the with the reference (p.9, lines 231-232)
Also, we fixed them as the reviewer pointed out (p. 19, lines 496-504).

Also, we use the terminology "organic soil" or "peat soil" in the comparison to other models as mentioned their references. But we emphasized that our approach is based on "organic-rich mineral soil" because of relatively low OC amount used in our validation (p.1, line 18 ; p.5, line 181; p.13, 342; p.18, lines 463-464)